# E-Cigarettes: A Disruptive Technology? An Analysis of Health Actors’ Positions on E-Cigarette Regulation in Scotland

**DOI:** 10.3390/ijerph16173103

**Published:** 2019-08-26

**Authors:** Heide Beatrix Weishaar, Theresa Ikegwuonu, Katherine E. Smith, Christina H. Buckton, Shona Hilton

**Affiliations:** 1MRC/CSO Social and Public Health Sciences Unit, University of Glasgow, Top floor, 200, Renfield Street, Glasgow G2 3AX, UK; 2School of Social Work & Social Policy, University of Strathclyde; Lord Hope Building, 141 St James Road, Glasgow G4 0LT, UK

**Keywords:** public health, tobacco control, electronic cigarettes, health policy, United Kingdom, Scotland, advocacy, evidence, policy debate

## Abstract

Concerns have been raised that the divisions emerging within public health in response to electronic cigarettes are weakening tobacco control. This paper employed thematic and network analysis to assess 90 policy consultation submissions and 18 interviews with political actors to examine the extent of, and basis for, divisions between health-focused actors with regard to the harms and benefits of e-cigarettes and appropriate approaches to regulation in Scotland. The results demonstrated considerable engagement in e-cigarette policy development by health-focused actors and a widely held perception of strong disagreement. They show that actors agreed on substantive policy issues, such as age-of-sale restrictions and, in part, the regulation of advertising. Points of contestation were related to the harms and benefits of e-cigarettes and the regulation of vaping in public places. The topicality, limitations of the evidence base and underlying values may help explain the heightened sense of division. While suggesting that some opportunities for joint advocacy might have been missed, this analysis shows that debates on e-cigarette regulation cast a light upon differences in thinking about appropriate approaches to health policy development within the public health community. Constructive debates on these divisive issues among health-focused actors will be a crucial step toward advancing public health.

## 1. Introduction

Electronic cigarettes (e-cigarettes) or electronic nicotine delivery systems (ENDS), devices that “heat a solution (e-liquid) to create an aerosol which frequently contains flavourants” [1] and often nicotine, have received increased public and policy attention since their invention in 2003. Their rapid proliferation and use by smokers and non-smokers has led to controversial public health and political debates, with proponents and opponents of e-cigarettes taking different positions on the potential implications of e-cigarettes and on the appropriate approach to regulation [2,3,4]. These controversies extend to organisations and individuals with an interest in public health that have, for decades, jointly worked to reduce the harms caused by tobacco through political advocacy. Indeed, the tobacco control community in the UK has been noted for its effectiveness in achieving research-informed policy change [5], with the UK topping an international ranking of tobacco control for consecutive years [6]. Analysts argue that this success is at least partially attributable to the ability of researchers, advocates, practitioners, policymakers and journalists to work collectively, despite not always sharing the same underpinning values or rationales [7]. Meanwhile, analysis of internal tobacco industry documents (obtained through litigation cases in the USA) has revealed that the tobacco company Philip Morris USA initiated “Project Sunrise” from 1994–2006 with the intention of undermining the effectiveness of the tobacco control community by exploiting differences of opinion [8]. The high-profile and seemingly profound nature of the disagreements surrounding e-cigarettes [9] suggests that political actors who previously unified their activities in efforts to improve public health are now increasingly perceived as divided in precisely the way Philip Morris USA hoped.

While the disagreement on e-cigarettes has been highlighted in previous publications [9,10,11], it remains unclear which specific actors disagree, why they disagree, and on what particular aspects of the debate they disagree. This paper is the first to examine the debate that occurred in the UK, specifically on the Scottish proposals for e-cigarette regulation, to shed light on the views and positions of health-focused policy actors. It particularly focuses on analysing agreement and disagreement among the public health community regarding the harms and benefits of e-cigarettes and the appropriate level of regulation. While the paper examines policy debates in Scotland, it includes UK and English actors (since they engaged with the Scottish policy consultation; the UK has devolved powers on health to its nations, which means that Scotland adopts its own health policy). The UK context lends itself to studying e-cigarette policy debates, as it is at the forefront of e-cigarette regulation, with many leading international experts influential in e-cigarette policy and debates [5,12,13]. E-cigarette use is higher and its advertising is more prevalent in the UK than in neighbouring European countries [14]. A specificity of Scotland is that, since political devolution in 1999, responsibility for many (though not all) aspects of e-cigarette regulation lie with the Scottish government and that Scotland has increasingly assumed public health policy leadership [15]. The analysis in this paper is likely to assist those working to protect and improve public health in countries around the world to understand e-cigarette debates in the UK. It may help to highlight potential areas of convergence as well as issues that need further reflection, thus supporting the development of more constructive conversations.

## 2. Materials and Methods

To shed light on a recent example of Scottish regulation of e-cigarettes, this article focuses on the Health (Tobacco, Nicotine etc. and Care) (Scotland) Act of 2016, one of the most comprehensive pieces of legislation on e-cigarettes passed in the UK [16]. With regard to e-cigarettes, the Scottish Act introduces age-of-sale restrictions for youth under 18 years of age; a tobacco and nicotine vapour product retailer register; and restrictions on advertising and promotion, including brand stretching, sponsorship, free distribution and nominal pricing. The development of the act is summarised in chronological order in Table 1. It is important to note that in parallel to the Scottish regulation, the Nicotine Inhaling Products (Age of Sale and Proxy Purchasing) Regulations of 2015 were adopted, introducing age-of-sale restrictions on e-cigarettes for those under 18 in England and Wales [17].

### 2.1. Overview

This study was undertaken as part of a larger mixed methods research project that combined (i) a documentary analysis of responses to the two consultations in preparation for the act and related websites, reports, briefings and other documentary data; (ii) a network analysis of the links between the websites of actors and their positions on elements of e-cigarette regulation, identified in the documentary analysis; and (iii) a thematic analysis of semi-structured interviews with a range of actors with an interest in UK e-cigarette policy, conducted after the Health (Tobacco, Nicotine etc. and Care) (Scotland) Act of 2016 was passed. In this paper, we focus on the analysis of the data for a subset of actors with a primary interest in health. The study was reviewed by and obtained ethical approval from the University of Glasgow College of Social Sciences Ethics Committee for Non-Clinical Research Involving Human Subjects (application number: 400150145).

### 2.2. Documentary Data Collection

Documentary data were derived from the 2014 Scottish government consultation on its draft paper “Electronic Cigarettes and Strengthening Tobacco Control in Scotland” and from the 2015 Scottish Parliament Health and Sports Committee consultation on the Health (Tobacco, Nicotine, etc. and Care) (Scotland) Bill. Both consultations provided an opportunity to identify key actors in the e-cigarette regulation debate. Responses were included in the sample if they derived from an organisational actor (as opposed to an individual), dealt with the regulation of e-cigarettes and were publicly available. Based on these inclusion criteria, 121 out of 266 consultation responses were selected for inclusion in the documentary analysis (Figure 1). In addition, key websites (including the Scottish Parliament’s website, which lists documents related to the policy process; the Scottish government’s website on the “Consultation on Electronic Cigarettes and Strengthening Tobacco Control in Scotland”; and the Scottish Parliament’s Health and Sports Committee’s website) were searched to identify other documents of relevance to the policy process, as well as key individuals and organisations involved in the process and their positions.

### 2.3. Identification of Health-Focused Actors

This study focused on examining consultation responses and other documentary data from organisations with a primary health focus. Health-focused actors were identified as organisations, charities or representative bodies whose work focused on tobacco control or other public health issues, the prevention and cure of specific diseases, health research or health service provision. Where it was not immediately clear whether an organisation could be labelled as health-focused, internet-based research was undertaken to assess its (or, if this was not possible, its UK/EU umbrella organisation’s) view on health issues and decide whether to label the respective organisation as health-focused. Through this process, 90 of the 121 organisations that engaged in the Scottish government consultation were identified as having a focus on health. Table 2 lists all types of organisations that submitted responses to the consultation process, differentiating between actors that were labelled as health-focused (*n* = 90) and those labelled as non-health-focused (*n* = 31).

### 2.4. Network Analysis

Consultation responses submitted by the 90 health-focused organisations were investigated using network analysis. Given its ability to investigate network structure and links between different actors, network analysis was deemed an appropriate tool to explore the extent and appearance of consensus and dissent between health-focused actors. Each organisation was identified as a node. Attribute data for each node were systematically extracted from the consultation responses and organisations’ websites, including characteristics with relevance to the research question. These included type of organisation, primary interest, geographic location, level of engagement in the policy process and the organisation’s position on each aspect of the proposed e-cigarette regulation. Six pre-set opportunities for engagement were treated as proxies for the level of actor engagement in the policy process (Table 3). The chosen events were identified during the documentary analysis as crucial opportunities seized by interested actors to engage in the process. A thorough search of all policy documents for attendance at or participation in the respective event by representatives of the 90 health-focused organisations allowed rating each organisation’s level of engagement. All data were numerically coded in an Excel spreadsheet.

Web links were used as proxies for the relationships between the health-focused actors and were identified using the web crawling software IssueCrawler (www.issuecrawler.net). IssueCrawler is a web network location and visualisation software that crawls specified sites and captures outlinks from those sites to other specified web sites. The interactor setting of IssueCrawler was used with the maximum crawl depth possible (three degrees of separation) to extract directed ties between network actors that were up to “three clicks away”. Manual web crawling was carried out for the small number of websites that were not accessible to IssueCrawler. Any organisations that did not have a website and could therefore not be crawled were included in the data file as having no links with any other actor in the network. All data were treated as non-valued data. All isolates were removed, and the strength of tie was set to be the same for all relationships before analysing the network. Network analysis was carried out using the software Gephi (version 0.9.1). In order to analyse the nature and structure of the network and to identify actors that had important positions within the network as well as any relevant links between network actors, centrality analyses using eigenvector calculations were conducted. The Fruchterman–Reingold algorithm and Gephi were used to display the network map.

Given the prominence of age-of-sale restrictions, the regulation of advertising and marketing and the regulation of vaping in public places in the debates, these policy elements were identified as the most significant and were selected as the focus for a more detailed network analysis. The sociograms were colour-coded to highlight the position of health-focused actors on each of these three policy areas.

### 2.5. Interviews

A list of political actors engaged in e-cigarette policy was developed from the documentary data, which was then used to purposively select and prioritise potential interviewees. Our aim was to obtain maximum variation in the sample of actors representing different types of organisations, different positions on e-cigarette regulation and different levels of engagement in the policy process [22]. Potential interviewees were approached by email or telephone invitation. A topic guide was developed that covered questions on reasons for engagement in the policy process; actors’ interests in, and positions on, the benefits, harms and regulation of e-cigarettes; actors’ use of evidence; actors’ efforts to shape e-cigarette policy; and their roles in the policy debate. The interviews typically lasted between 45 and 60 min and were either conducted face-to-face or by telephone, depending on participants’ preferences. All interviews were recorded and transcribed verbatim. Transcripts were read and reread by the authorship team prior to developing and agreeing on a thematic coding frame. Each transcript was coded and analysed using iterative comparisons and the recoding of emergent themes. The principle of the constant comparative method was used to help identify explanations of patterns across the data [23]. A total of 25 semi-structured interviews were conducted between October 2016 and August 2017. The interviews were conducted with a variety of actors who had an interest in Scottish e-cigarette regulation, including representatives of third sector/civil society organisations with a specific interest in health (*n* = 5), the academic sector (*n* = 4), health service providers (*n* = 4), health service/National Health Service (NHS) organisations (*n* = 3), representatives of manufacturers of traditional cigarettes (*n* = 2) and e-cigarettes (*n* = 2), representatives of local authorities with a health remit (*n* = 1), government authorities with a health remit (*n* = 1), the retail sector (*n* = 1), manufacturers of pharmaceuticals (*n* = 1) and vapers (*n* = 1). The focus of the qualitative analysis for this paper was on the interviews with actors that represented organisations with an explicit focus on health (*n* = 18). Data from interviews with actors representing organisations that had no explicit health focus (*n* = 7; i.e., manufacturers of traditional cigarettes, e-cigarettes and pharmaceuticals; the retail sector and vapers’ groups) are only drawn on below sporadically to highlight external perceptions of the network, overlaps in views or disagreement. In these cases, it is explicitly made clear that other interview data are being referred to. Interview participants were guaranteed anonymity, and therefore when reporting interview data, we use a generic descriptor that provides only the professional context of the speaker. We cannot provide further details on the interviewees. Given that some representatives of organisations with an interest in health, including individuals that were interviewed as part of the project, were uncomfortable with the term “stakeholder” because they felt that it suggests that tobacco companies have a legitimate interest/stake in health policy, it was decided to use the term “(political) actor” in the project to refer to political actors with an interest in e-cigarettes and to avoid the term “stakeholder”.

Some supplementary content analysis of the documentary data was undertaken and used to triangulate the network as well as the interview analysis.

### 2.6. Limitations of the Study

The study had a number of limitations. First, the case study focused on the policy debate on e-cigarettes in Scotland. While the UK is perceived to be at the forefront of e-cigarette debates [6], conclusions about other political contexts must remain tentative. Differences in regulation regarding the nicotine content of e-cigarettes in the EU and the USA mean that e-cigarette debates might differ significantly between the USA and the case investigated in this paper. Second, the selection of interviewees with specific views on the policy might mean that some topics received more attention in the interviews than others. Using the documentary data analysis, however, considerable efforts were made to reach a sample that included a wide range of individuals who represented different organisations, sectors and positions on e-cigarette regulation. Third, the links that were detected using web crawling might indicate permanent, structural links between the organisations that engaged in the policy debates, rather than links that were specific to the context of e-cigarette policy. In order to respond to this limitation, the study used data triangulation with interviews to explore the nature of political actors’ engagement in the UK policy debates to provide a more comprehensive picture and achieve a better understanding of the complexity of political engagement.

## 3. Results

### 3.1. Engagement of Actors Focused on Health

Of the 90 health-focused organisations that engaged in the e-cigarette policy debates, almost half had an explicit primary interest in health (*n* = 42). This included organisations representing different aspects of the NHS, disease-specific lobby organisations and health professionals. An additional eight organisations’ primary interest was specifically tobacco control. Other actors included academic groups and organisations representing the interests of consumers, young people or others. Sixty-five organisations were located in Scotland, with 48 being local and 17 Scottish national organisations, 2 were in England, and one was in Wales. Twenty-one were UK-wide organisations, and one was a European organisation. Most actors (*n* = 62) had engaged in only one policy engagement opportunity, whereas two organisations (NHS Health Scotland and Action on Smoking and Health (ASH) Scotland had seized all six opportunities to engage in the policy process. (In order to distinguish between the different Action on Smoking and Health (ASH) organisations that operate in the UK, we use the term ASH Scotland to refer to the Edinburgh-based ASH organisation and ASH (in England) to refer to the London-based ASH organisation.)

A detailed analysis of the structure of the network of all organisations with an interest in public health showed that multiple connections existed between the different types of organisations (Figure 2). In particular, civil society and NHS organisations were well connected, as were NHS organisations and local authorities. Professional bodies were well connected with other professional bodies as well as with civil society organisations, whereas pharmacies were almost exclusively linked with professional bodies, rather than with any other organisational type. The few academics in the network were well connected to civil society organisations and also displayed a few links with NHS organisations. Interestingly, the analysis of the interview data suggested that actors with a focus on health perceived themselves as a bounded coalition, thus supporting the findings of connectivity identified through the network analysis. Several interviewees referred to “the community” when referring to the group of actors with a focus on health.

NHS Health Scotland, Health Information Services/NHS 24, ASH Scotland and, to a lesser degree, ASH (in England) emerged as particularly central actors with high betweenness centrality (Figure 2), i.e., as actors who were connected to other actors that were also highly connected. ASH Scotland’s high eigenvector centrality scores matched the finding that the organisation had seized several opportunities to actively engage in the policy process: they were perceived by many interviewees as prominent lobbyists on Scottish tobacco control policy and seemed to be important actors in the network as well as in the debate on e-cigarette regulation.

### 3.2. Disagreement between Health-Focused Organisations

While both the network as well as the thematic analysis showed that health-focused actors were well linked with each other, the interviewees also reported that considerable disagreement existed among the community with regard to e-cigarettes. One academic interviewee, for instance, highlighted:
“There is a lot of disagreement about whether electronic cigarettes are positive or negative for public health. There is a huge amount of disagreement.”

Different interviewees noted that the issue was “divisive” (two researchers, one health professional) and that debates were “quite polarized” (health professional) and “antagonistic” (researcher). Highlighting the divisions between health-focused actors, interviewees reported that those engaged in e-cigarette policy showed behaviour that was viewed as “a bit tribal” (third sector policy professional). A researcher said jokingly:
“It’s a bit like George W. Bush’s, kind of, ‘You’re either with us or against us.’”

Other interviewees referred to the two “duelling” (health service representative) letters that in 2014 had been sent to Margaret Chan, Director General of the World Health Organization (WHO), by members of the public health community, outlining arguments for and against the regulation of e-cigarettes [24,25]. Interestingly, disagreement among actors focused on health was not only described by members of the community themselves but also seemed apparent to potential opponents of health advocates, e.g., those with commercial interests. A transnational tobacco company representative, for example, highlighted that health-focused actors with an interest in e-cigarette debates could generally be divided into those who “appear determined to prevent their spread and their uptake” and the “side of the public health community which supports e-cigarettes”.

Interviewees highlighted that disagreement even existed between high-level organisations that were formally linked to each other. Groups “that you would’ve thought should have a similar line” (health professional) were reported as disagreeing on e-cigarette use and regulation. Such disagreement was also recognised by non-health actors that engaged in the policy debates and was perceived as confusing, as the following quote by a representative of a pro-e-cigarette campaign group illustrates:
“It just seems bizarre that you can have people at Public Health England with the “right” view of vaping completely opposed to the Faculty of Public Health with almost polar opposite views of vaping. And then when you look at the fact that the Faculty of Public Health is actually a faculty within the Royal College of Physicians which is pro-vaping, you know, the picture just becomes so complicated.”

### 3.3. Contrast to Previous Agreement within the Public Health Community

The interview analysis suggests that dissent developed fairly recently and was perceived as specific to the context of e-cigarette policy debates. Representatives of organisations with a health focus frequently mentioned that the sense of controversy was in stark contrast to the previous unity that they had perceived to exist on other tobacco control issues: one health professional said:
“It’s usually quite rare on a public health topic that we don’t all generally agree.”

Reflecting on their involvement in tobacco control over the last 20 years, one health professional concluded that they had “never known an issue to cause the divisiveness that this has caused”. Similarly, a health service representative highlighted that e-cigarettes had “absolutely divided the community which was so united”. In this context, the previous joint opposition against transnational tobacco corporations was identified as a crucial unifying factor. Interviewees observed that the opposition against transnational tobacco corporations had disintegrated in the context of e-cigarette debates, resulting in increasing disagreement between public health organisations. An academic, for example, highlighted:
“Now it’s public health fighting among itself instead of what we used to do, which is we knew who the enemy was. The enemy was the tobacco industry and that has, you know, the debate has shifted.”

Similarly, a health charity representative reflected on the previous unity in judgement, which, according to her opinion, had been disrupted by e-cigarettes:
“Because up until now, we had a clear ‘good’ and ’bad’, ’good’ versus ’evil’, ’black’ versus ’white’, ’tobacco - bad’, et cetera. E-cigs have thrown that into significant confusion.”

### 3.4. Emotive Debates and the Role of High-Profile Actors

The interview data also revealed that e-cigarettes and their regulation were perceived as “a very, very emotive subject” (representative of a UK health charity), with interviewees reporting that interactions between different public health actors had been “vicious […], insulting […] and rude” (academic), that people had at times been “really quite nasty” (representative of a third sector organisation), and that those who had been at the receiving end had been “shocked” (public health professional; health charity representative), “horrified” (public health professional), “quite upset” (representative of a third sector organisation) and “very distressed” (academic). Interviewees reported that conflicts had been fought out publicly and had undermined their relationships. A representative of a UK health charity reported:
“[It] has actually split the tobacco-control field and has actually divided what you might consider professional friendships. People who have been like that [crosses fingers] on tobacco control are all of a sudden completely apart. And it’s become incredibly controversial. It is a situation where all of a sudden incredibly experienced and well-respected global academics are publicly challenging one another and often falling out on what is becoming an increasingly partisan basis.”

Some interviewees reported that a few vocal and high-profile individuals who took strong positions had contributed to the polarisation of the debate. In attempts to stress that they themselves had not done the same, several interviewees reported that they cultivated good relationships with other actors with a focus on health, were positioned “in the middle” (health service representative) and took moderate positions on e-cigarettes and their regulation. In a similar attempt to highlight agreement in the policy debates, some interviewees stressed that disagreement was not really fundamental and that opinions were “not as far apart as some people seem to think” (researcher). Instead, they claimed that any controversies concerned rather peripheral issues and were mainly due to the different emphases that were placed on different aspects of the debate.

### 3.5. Consequences of Disagreement

Despite trying to stress areas of consensus, interviewees admitted that existing divisions precluded actors from working together. As several interviewees recalled, this, in turn, meant that no consensus statements were developed on e-cigarettes. One public health professional said:
“I don’t remember seeing any sort of, even attempt, to have a consensus statement between public health groups.”

The lack of consensus among the public health community was broadly viewed as negative, as it prevented advocacy organisations from building on past coalitions and developing joint positions on issues to increase their voice, political impact and visibility. Importantly, the existing dissent on e-cigarettes was perceived as decreasing the public health community’s ability to impact the policy process, with interviewees reporting that policymakers had found it difficult to make decisions on the regulation of e-cigarettes in the light of apparent controversies. A health professional highlighted that it is “impossible to come up with a proper regulation […] when so many important groups have a different opinion on it”. Some interviewees even attributed the varying approaches to regulation in different UK legislatures and health boards to the dissent among the public health community and the lack of a clear message on the part of health organisations. Concerns were also voiced in relation to the confusion among the general public about the harms, benefits and appropriate regulation of e-cigarettes.

Interviewees also stressed that e-cigarette debates had “hijacked the tobacco control agenda” (representative of third sector organisation) and had drawn a lot of attention away from other tobacco control issues, which in turn did not receive the consideration they deserved and needed. The following academic expressed strong concerns about this:
“The e-cigarette issue is so divisive that a lot of debate is directed towards that. We have a lot of meetings, a lot of discussion, and I think it takes away from pursuing evidence-based tobacco control, things like mass media, tax, […] smoke-free, smoking cessation services. I think that all of those things have been damaging because we’ve spent so much time fighting about e-cigarettes.”

On this, interviewees reported that the focus on e-cigarettes had taken funding and attention away from other areas of tobacco control. A health service representative, for example, highlighted:
“[The e-cigarette debate meant that] loads, millions of pounds [were put] into research that could have, and would have, gone into tobacco. It’s been a diversionary tactic.”

One health service representative reflected on the relative importance of e-cigarettes and on people losing perspective, stating:
“[E-cigarettes] are just a tiny bit of the pie, and everyone’s forgotten that there’s a whole cake sitting on the rest of the table. […] But all of the stuff we could be trying to talk about and we could be trying to get innovative policy on is languishing because there’s so much time being spent on e-cigarettes.”

### 3.6. Agreement and Disagreement on the Appropriate Level of Regulation

The majority of health-focused organisations expressed an opinion on all elements of the e-cigarette regulations proposed by the Scottish government when submitting a consultation response. Response rates ranged from 66% on the introduction of fines for failing to verify age to 94% on setting a minimum age of sale at 18 (Table 4).

In the context of the very clear sense of division that interviewees described, the documentary analysis of consultation responses suggests, somewhat surprisingly, a high degree of consensus among health-focused actors on many elements of proposed e-cigarette regulation (Table 4). The interview data suggest that there was uniform agreement that e-cigarettes are a less harmful form of tobacco consumption than traditional cigarettes.

When focusing on the three main areas of regulation—age-of-sale restrictions, regulation of e-cigarette advertising and marketing and vaping in public places—agreement seemed to be particularly high regarding the first and, though to a lesser degree, second regulatory proposal. As the network analysis illustrates, most health-focused actors agreed that the sale of e-cigarettes to those under 18 years old should be forbidden. The overwhelming majority of health-focused actors (*n* = 84, 93.3%) supported respective regulatory proposals, with only 5 organisations (5.5%) taking no clear position on the issue and one (1.1%) organisation opposing age-of-sale regulations (Figure 3a). Regarding the advertisement and promotion of e-cigarettes, there was still considerable agreement, though slightly less than that regarding age of sale and with point-of-sale advertising emerging as a particularly divisive issue. While most health-focused organisations (*n* = 76, 84.4%) supported restrictions on e-cigarette advertising and marketing, 10 organisations (11.1%) took no clear position, and four organisations (4.4%) did not support regulation, including ASH (in England), who was a leading policy actor (Figure 3b). The interview data provided further evidence on controversies that existed with respect to appropriate advertising regulations. The dividing lines seemed to be between a desire to inform potential consumers and communicate with smokers about switching to less harmful products and the wish to prevent tobacco companies from advertising an addictive product to non-smokers and young people. When arguing against the stringent regulation of advertising, interviewees highlighted that it was “important to get products known to […] potential consumers and to communicate information on risks and benefits of different products” (academic) and that advertising would allow those providing e-cigarettes “to speak to the smokers, get them to quit” (academic). Point-of-sale advertising was highlighted as a particularly important way to allow potential consumers “to know how to use the devices, […] try them in a vape shop, […and…] get reliable information” (academic). Interestingly, often the same individuals that provided arguments for allowing some forms of advertising also acknowledged that safeguards had to be put into place to restrict the advertising of e-cigarettes to non-smokers. When putting this argument forward, interviewees frequently referred to well-known tobacco industry strategies to promote their products to vulnerable groups and the difficulties in enforcing the necessary regulation. One academic said the following:
“The tobacco industry will use […] advertising to promote uptake by children ‘cause they wanna grow their market as much as possible and we don’t want that. So, it’s legitimate to have restrictions on advertising to children but then very difficult to impose or police.”

One interviewee (an academic) provided a neat summary of the dilemma that health-focused actors seemed to face when trying to develop an opinion about e-cigarette advertising:
“That’s incredibly complex because the evidence isn’t really there. […] You want to be able to communicate with smokers, but you don’t actually want to get a whole load of non-smokers onto e-cigarettes.”

While the analysis of consultation responses suggested a fairly high level of agreement between health-focused actors with regard to age-of-sale regulation and restrictions on advertising, the issue of banning vaping in public places was more divisive. Only 54 organisations supported banning vaping in public places (60.0%), 25 organisations did not take a clear position (27.7%), and 11 (12.2%) organisations opposed such regulation (Figure 3c). Interestingly, England-based pharmacies and health professionals, health charities and all ASH organisations (in Scotland, England and Wales) did not support a ban of vaping in public places.

Contentious debates also seemed to emerge around the harmfulness of nicotine as a drug that was inhaled when using e-cigarettes. Many actors concluded that smokers, who were the intended audience for e-cigarettes, would benefit from switching to e-cigarettes, which allowed them to ensure nicotine intake while avoiding consuming the other harmful substances contained in traditional cigarettes. Others stressed concerns about continuous nicotine consumption. There was also some debate about the scope of e-cigarettes to increase cessation and help smokers quit. While many focused on the potential of e-cigarettes in helping smokers stop smoking, it was also highlighted that they had to be considered as part of a much broader tobacco control strategy. In fact, a number of interviewees voiced doubts as to whether e-cigarettes would have a substantial impact on ending the smoking epidemic. As the following quote by one third sector professional illustrates, these interviewees stressed that some e-cigarette supporters, notably vapers, were prone to exaggerate the likely impact of e-cigarettes:
“I mean, there are people who think that they’re some kind of a magic bullet which will solve the tobacco epidemic….”

Several public health advocates cautioned against e-cigarettes because they hypothesised that they might provide a gateway into traditional smoking for non-smokers and particularly for young people, encourage dual use and contribute to the renormalisation of smoking. Arguments here centred on the imitation of smoking-like behaviour, public visibility and the subsequent potential acceptance of e-cigarettes. These arguments were particularly highlighted when talking about regulations of advertising and promotion and vaping in public places. E-cigarette marketing was compared to marketing alcopops, which was perceived as the promotion of a product that would be “appealing to youngsters […] where you are encouraging experimentation” (health professional). A government representative voiced “fears that tobacco manufacturers will hijack this technology and use it to nurture and gain new smokers […and to…] get kids hooked on nicotine through this legal means, and then […] they can sort of move to become smokers”. In this respect, the lack of safety and the addictive potential of nicotine were also important areas of concern.

### 3.7. Reasons for Disagreement

The analysis suggested that a number of factors shaped the disagreements on e-cigarettes, including (i) the topicality, changing nature and popularity of the issue; (ii) the lack of evidence on e-cigarettes; and (iii) the underlying values and views that influenced advocates’ positions.

Interviewees highlighted that e-cigarette debates had taken place in a rapidly changing context, as new products were developed, companies merged, and the market moved “at a huge fast pace” (health professional). The dynamics of the market posed challenges to advocates, who reported that they struggled to keep pace with what seemed like a moving target. The rapid proliferation of e-cigarettes and the huge media interest in them were highlighted by several interviewees as crucial factors that shaped public and regulatory debates. Media reporting was often perceived as unhelpful. Advocates said that they had been approached by the media to comment on e-cigarettes and reported that they had perceived journalists as trying to set up a controversial debate and backing them “into a position that we wouldn’t favour” (representative of a third sector organisation). While health actors reported that they tried to refrain from promoting simplistic viewpoints or making statements that were not backed up by sufficient evidence, they acknowledged that this was difficult given the media’s desire for catchy headlines. Interviewees complained that “media reports […] tend to focus on the extreme aspects of the kind of overarching headlines, rather than the detail” (health professional), and that the lack of detail in media reporting had increased controversies among those who had been cited.

Interviewees reported that responding to information requests about e-cigarettes, agreeing on public statements and developing joint positions on policy proposals had been particularly difficult because of the lack of evidence on the harms and benefits of e-cigarettes. One health charity representative postulated that any strong positions on the subject were not justified:
“Anyone that takes a really strong stance on e-cigarettes is wrong, because the evidence is not there yet to take a strong stance.”

Unsurprisingly, inconclusive evidence seemed to be even more of a problem for organisations with a commitment to evidence-informed policy. Representatives of organisations that were focused on research highlighted that their positions had to be based on the best available evidence. They were concerned that inconclusive evidence had caused problems in developing, agreeing on and communicating a clear message to policy audiences and also in legitimising their engagement in the debate: one health charity representative said:
“One of the challenges of an organisation like ours is that everything we have a policy on must have an evidence base. The very essence is: What is the data? What is the evidence? What firm conclusions…?”

In light of the inconclusive evidence, it might not be surprising that the interview data, in fact, provided some indication that the positions of health-focused actors on e-cigarette policy were not necessarily and primarily influenced by the available evidence. They were also informed by personal experience, normative ideas and views on the primary aims of public health policy and deep-rooted values about public health. An academic, for example, referred to “entrenched views and values” (academic) that influenced the policy debates. Other interviewees mentioned firmly held beliefs, values and views which they perceived as inflexible and deeply rooted in people’s self-image and their previous experiences of working in, and advocating for, public health. Reflecting on the origin of these values, selected interviewees reported that the tobacco industry’s history of deception had caused immense reluctance on the part of some members of the public health community to engage with commercial actors producing e-cigarettes or consider anything that the industry promoted. A health service representative highlighted:
“Some people […] feel very, very strongly. And it’s all, it‘s very much tied in with their, you know, personal feelings around the tobacco industry.”

Interviewees who were less opposed to working with companies manufacturing e-cigarettes (which often also manufactured traditional cigarettes) perceived this opposition to transnational tobacco companies as a barrier to reacting flexibly to the quickly changing e-cigarette market, to having constructive policy debates, to considering opposing views and the evidence that informed them, to building coalitions and eventually to developing and implementing pragmatic policy solutions. The same interviewees often argued that those who were dealing with smokers on a regular basis and were thus confronted with the harms caused by traditional tobacco were more likely to be in favour of e-cigarettes as pragmatic and potentially life-saving quit aids. They contrasted their own approaches to harm reduction with those of actors who were not confronted with the suffering of smokers and thus less likely to support e-cigarettes as a means to reduce harm. However, this hypothesis was not supported by the consultation data, which showed that it was equally likely for health service providers, including representatives of health professionals, and for other health-focused actors to support high levels of regulation, including regulation on e-cigarette advertising and marketing as well as restrictions on use in public places.

## 4. Discussion

Drawing on the Health (Tobacco, Nicotine etc. and Care) (Scotland) Act of 2016 as a recent case of e-cigarette policy development within the UK, this paper is the first to provide an in-depth analysis of debates on e-cigarette regulation. The paper not only sheds light on the views and positions of key actors with an interest in the development of e-cigarette policy but highlights common ground as well as crucial areas of contestation and issues that might require further attention by those interested in advancing public health. The analysis shows that representatives of health-focused organisations, including third sector/civil society organisations, local authorities, organisations representing the health service and health service providers, academic institutions and others, had a strong interest and made extensive efforts to actively engage in the policy debates. It also shows that organisations that, in the context of previous tobacco control policy debates, had pursued common goals and aligned key messages [26] were divided with regard to their assessment of the harms and benefits of e-cigarettes as well as on aspects of e-cigarette regulation, including vaping in public places and, to a lesser degree, the promotion of e-cigarettes. This resulted in an inability to build coherent coalitions and to jointly advocate for specific policy approaches.

A detailed analysis of the disagreements also indicates that most health-focused organisations supported age restrictions on e-cigarette sales and were in some agreement on the regulation of e-cigarette advertising. This suggests that political actors could have chosen to focus on developing more unified positions around age-of-sale and advertising restrictions and may have missed an opportunity for building an advocacy coalition to influence policy. Our data suggest that, instead, actors frequently emphasised the areas of policy on which they disagreed. While our analysis outlines that there were differing views among health-focused actors about aspects of e-cigarette regulation and that this contributed to a sense of division, the question is why political actors with an interest in health did not focus on common ground. Comparing this case study of e-cigarette policy to other issues of tobacco control policy, we can see that there have often been different views within public health about specific aspects of tobacco control regulation [7,27,28], but also that these disagreements have not undermined tobacco control coalitions in the same way that e-cigarette legislation appears to have done. It is important, therefore, to consider why debates about e-cigarettes have proven so divisive.

Our paper provides some indication that in the context of Scottish e-cigarette policy, the preoccupation with areas of disagreement was exacerbated by the following closely related factors: (i) the lack of conclusive evidence on the harms and benefits of electronic cigarettes; (ii) the lack of evidence regarding the effects of regulation; (iii) differing underlying views on appropriate approaches to health policy development; (iv) the sudden emergence and rapid proliferation of e-cigarettes and the fast pace of technological change (which meant emerging evidence was often incomplete or out of date and sometimes conflicting); and (vi) the huge media interest in the e-cigarette debate. This media attention resulted in frequent requests for academics and political actors to make public statements. As a consequence, actors with an interest in the debate often, deliberately or unintentionally, positioned themselves in more polarised positions, which sometimes seemed influenced primarily by views and opinions rather than evidence. The lack of conclusive evidence, partially caused by a lack of rigorous clinical trials on e-cigarette use and missing evaluations of policy interventions, might have made actors more inclined to make statements that were influenced by personal opinions. It could be argued that this inclination might also have previously contributed to public dissent among health-focused actors about a report published by Public Health England [29]. The report, which claimed that e-cigarettes are 95% less harmful to one’s health than normal cigarettes, was criticized by major international health journals for basing its claims on a study that had major methodologic limitations, for not acknowledging the potential conflicts of interests of those involved in judging the harms of e-cigarettes and for failing to meet basic evidentiary standards [2,30,31]. The media representation of actor positions on other public health policy debates, such as the case of the UK Soft Drinks Industry Levy, provides another example of similar dynamics in light of inconclusive evidence [32]. Here, Hilton et al. [32] found that there were inconsistencies in the way actors communicated their support for the policy in public statements, which arose from insufficient clarity on the nature of the problem and conflicting policy priorities; leaving the actor’s degree of support open to interpretation.

In addition to the inconclusive evidence on the harms and benefits of e-cigarettes and the effects that would be expected in case of regulation, views on transnational nicotine and tobacco companies seemed to hinder the development of consensus among health-focused actors. The fact that these companies had become major actors in the e-cigarette market, i.e., in the market of products that some actors perceived as effective quit aids, meant that some actors felt that they needed to re-consider their views about tobacco companies’ engagement in harm reduction, with some concluding that tobacco companies should be treated as legitimate stakeholders. Other actors, however, continued to oppose tobacco industry engagement in health policy and were concerned that e-cigarettes provided the long-sought-after industry opportunity to “address tobacco industry delegitimisation” and “ensure the social acceptability” of tobacco companies [8]. Their underlying views on the need to exclude this industry from e-cigarette debates in order to protect public health took precedence over seeking consensus with other health-focused actors on the appropriate level of e-cigarette regulation. This situation seemed to result in dissent and was in contrast to the previously uniform reluctance of most health-focused organisations to engage with any tobacco industry-affiliated actors [26]. Shifting positions on tobacco industry legitimacy seemed to result in the loss of a sense that tobacco control was united by its common “enemy”, a previously important glue and coalescing force that had held the public health community together in tobacco control debates [7,26,33]. It also meant that dissent evolved within the health community rather than between actors with an interest in health and actors with an economic interest, much as Philip Morris USA had tried to achieve via “Project Sunrise” between 1994 and 2006 [8]. Negative consequences included the disengagement of experts from the debates and the disruption of professional friendships and collaboration, but more importantly, an inability to build advocacy coalitions and jointly lobby on a specific approach to regulation. This helps explain why, although most interviewees positioned themselves as occupying a “middle ground” on e-cigarettes, few felt that the polarised nature of the debates allowed voices from the middle ground to emerge or be heard. Interviewees generally agreed that this was likely to contribute to confusion and uncertainty among public and policy audiences.

The analysis provides some indication that divisions were also influenced by underlying priorities regarding how to improve public health. It suggests that those with a harm reduction viewpoint were more likely to focus on supporting individual smokers in quitting and thus opposed the stringent regulation of e-cigarettes, whereas those with a stronger focus on the social and political determinants of health and the prevention of smoking uptake, notably among children and young people, seemed more likely to be critical of e-cigarettes and the marketing strategies of transnational tobacco companies and more likely to be in favour of stringent regulation.

Our study highlights a number of research areas that could usefully contribute to our understanding of e-cigarette and public health policy. First and foremost, comparative research could help to develop a better understanding of why e-cigarette debates are experienced as, and perceived to be, so divisive compared to other issues of tobacco regulation in which there have been differing views within public health. Such research could investigate the reasons for disagreement identified in this case study in more detail, including health-focused actors’ views on the engagement of tobacco and nicotine companies in e-cigarette policy, or the high media interest and dynamic nature of a policy issue. Further analysis of how evidence was understood, framed and employed within the debate on Scottish e-cigarette policy could help illuminate our understanding of the role of evidence in policymaking. In order to shed light on any vested interests among health-focused actors and the potential impact such interests have on health policy debates, it would be valuable to examine conflicts of interests among actors who engage in e-cigarette debates and investigate how health-focused actors with a conflict of interest (e.g., some academics, actors who receive commercial sector funding, pharmacists or other health professionals whose profits at least partly depend on e-cigarette sales) frame the debates. Finally, research that examines other, including more recent, cases of e-cigarette policies could help to establish whether any coalitions emerge around areas of consensus and could thus provide important insights into whether, and if so, how, public health actors are beginning to form alliances and promote consensual positions in e-cigarette policy debates.

## 5. Conclusions

The 2016 Scottish case study presented in this paper shows that actors with an interest in e-cigarette policy struggled to constructively respond to a lack of evidence and evident disagreements on the harms, benefits and appropriate approaches to e-cigarette regulation. Our analysis also suggests that some opportunities for building consensus, particularly with regard to age-of-sale regulations, may have been missed. The paper illustrates how the previous unity among public health actors on tobacco control issues broke down to some extent as a result of these struggles. The case study also shows that public displays of division and conflict were perceived by public health actors to have had detrimental effects on tobacco control in terms of the community’s credibility and policy influence.

Debates on e-cigarettes have developed considerably since the adoption of the Health (Tobacco, Nicotine etc. and Care) (Scotland) Act. Recent reports have suggested that some degree of agreement has formed among the public health community with regard to the usefulness of e-cigarettes in helping smokers quit and reducing the harms of smoking. The National Institute for Health and Care Excellence (NICE) guidance on e-cigarettes [34], a Scottish consensus statement that was published by Health Scotland and endorsed by various Scotland-based third sector, research and health service organisations [35], as well as the ASH report “Smoking Still Kills”, which was endorsed by over 120 organisations [36], provide evidence of consensus on this aspect of e-cigarette use. Other reports published in recent years have shown that health-focused actors have begun to reflect on some of the issues that have been identified as controversial in this paper, including the need for building a solid evidence base on the harms and benefits of e-cigarettes and on the effects of e-cigarette regulation [36,37]; the need to strike a balance between seizing the potential of e-cigarettes in helping smokers improve their health and minimising the risk of e-cigarette uptake by non-smokers, particularly young people and children [36,37]; and the special role of tobacco companies in the e-cigarette debates [38]. Constructive and respectful debates among those with an interest in advancing public health on these divisive issues will be a crucial step toward positive collaboration and creating a uniform public appearance. While disagreement about key issues continues to exist, those aiming to advance public health policy might want to highlight aspects of e-cigarette use and regulation they agree on in order to promote public health.

## Figures and Tables

**Figure 1 ijerph-16-03103-f001:**
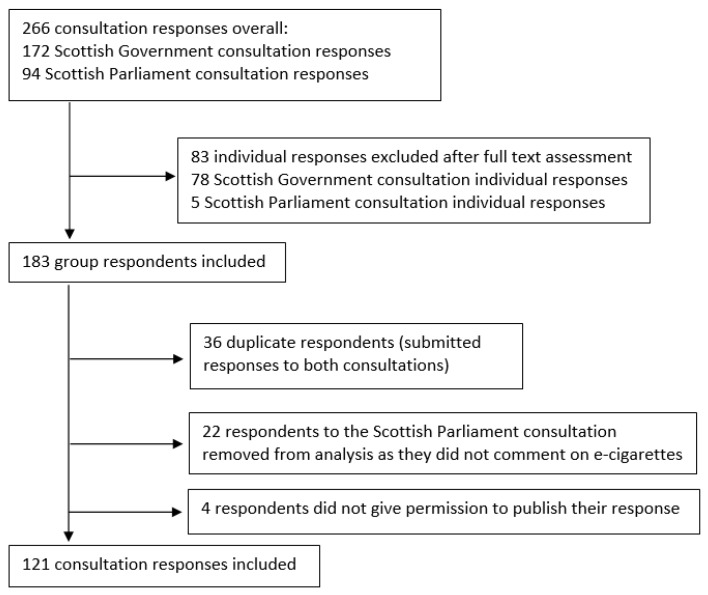
Flowchart of selection of consultation documents.

**Figure 2 ijerph-16-03103-f002:**
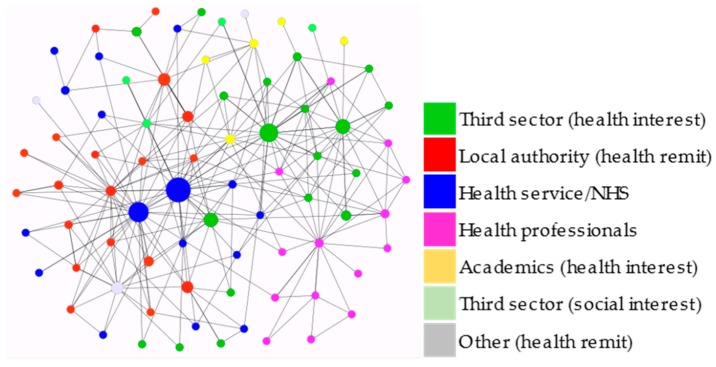
Network diagram of connections between health-focused actors (nodes are sized by betweenness centrality).

**Figure 3 ijerph-16-03103-f003:**
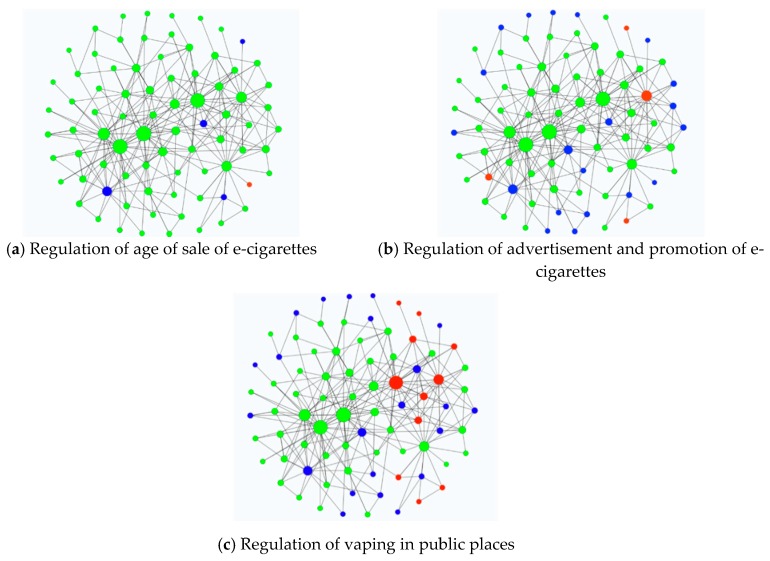
Network diagrams illustrating the position of health-focused actors on proposed policies, sized according to degree centrality. Colour-coding represents health-focused actors’ positions on each policy: green = supportive, red = not supportive, blue = unclear or missing data.

**Table 1 ijerph-16-03103-t001:** Chronology of the development of the Health (Tobacco, Nicotine etc. and Care) (Scotland) Act of 2016.

Date	Event
10 October 2014	Scottish Government launches a consultation on “Electronic Cigarettes and Strengthening Tobacco Control in Scotland” [18].
October 2014–April 2015	Consultation is open for submissions.
October 2014–April 2015	Meetings between Scottish Government and several political actors to consult on Scottish Government’s plans to adopt regulation of e-cigarettes.
May 2015	Scottish Government releases its report on the consultation on e-cigarettes and tobacco control in Scotland and its response to the consultation [19].
4 June 2015	The Health (Tobacco, Nicotine, etc. and Care) (Scotland) Bill is introduced in the Scottish Parliament.
Summer 2015	The Health and Sports Committee (designated lead parliamentary committee) consults with a range of experts on the bill, including representatives from the commercial sector, third sector/civil society, e-cigarette user groups, local authorities and health professionals [20]. The Health and Sports Committee also engages in a wider public consultation through the use of an online survey, Facebook, youth events and video blogs [20].
9 November 2015	The Health and Sports Committee publishes a Stage 1 Report on the Health (Tobacco, Nicotine etc. and Care) (Scotland) Bill, which demonstrates clear support for the bill [21].
November 2015–February 2016	Several rounds of amendments in the Scottish Parliament.
3 March 2016	The Health (Tobacco, Nicotine etc. and Care) (Scotland) Act of 2016 is passed in the Scottish Parliament.
6 April 2016	The Health (Tobacco, Nicotine etc. and Care) (Scotland) Act of 2016 receives Royal Assent [16].

**Table 2 ijerph-16-03103-t002:** Actors who submitted responses to the consultation process.

Focus	Type of Organisation	Number
Health	Third sector/civil society organisations with a specific interest in health	23
Health	Local authorities with remits in health (e.g., health improvement, consumers and environment)	21
Health	Health service/National Health Service (NHS) organisations	16
Health	Organisations representing health professionals	16
Health	Academic groups focused on health issues	6
Health	Third sector/civil society organisations with interests in broader health/social issues (e.g., health equality, children, sports and social care)	4
Health	Other organisations with remits in health (e.g., public body focused on violence reduction)	4
	*Total Health-Focused Actors*	*90*
Non-Health	Manufacturers of traditional cigarettes	9
Non-Health	Retail organisations	6
Non-Health	Manufacturers of e-cigarettes	4
Non-Health	Manufacturers of pharmaceuticals	2
Non-Health	Smokers’ and vapers’ rights groups with known links to the tobacco industry	2
Non-Health	Other commercial actors (e.g., advertising industry)	1
	Other organisations without a health focus (e.g., research organisation with known links to the tobacco industry)	7
	*Total Non-Health-Focused Actors*	*31*
Health and non-health	Total of all organisational actors submitting responses	121

**Table 3 ijerph-16-03103-t003:** Opportunities for actor engagement in the policy process: six levels of engagement.

1. Submission of a written response to the Scottish Government consultation process.
2. Attendance of a meeting with Scottish Government officials as part of the consultation process.
3. Attendance of a ministerial working group meeting prior to the publication of the first draft of the Health (Tobacco, Nicotine etc. and Care) (Scotland) Bill.
4. Submission of a written response on the e-cigarette aspects of the draft Health (Tobacco, Nicotine etc. and Care) (Scotland) Bill to the Scottish Parliament’s Health and Sports Committee consultation.
5. Provision of oral evidence to the Scottish Parliament’s Health and Sports Committee, the Scottish Parliament’s Finance Committee or the Scottish Parliament’s Delegated Powers and Law Reform Committee on the draft Health (Tobacco, Nicotine etc. and Care) (Scotland) Bill.
6. Attendance of a ministerial working group meeting after the publication of the first draft of the Health (Tobacco, Nicotine etc. and Care) (Scotland) Bill and prior to the adoption of the bill in the Scottish Parliament.

**Table 4 ijerph-16-03103-t004:** Overall agreement and disagreement with elements of e-cigarette regulation that were suggested in the consultation by health-focused actors. Data extracted from consultation responses.

Proposed Regulation	Response Rate ^1^	# Actors Agree	# Actors Disagree
		*n*	*n*
**The Minimum Age of Sale for E-Cigarettes should be set at 18.**	**94%**	**84**	**1**
AoS regulation should apply to all products, not just those containing nicotine.	71%	60	4
AoS regulation offence should apply to both retailer and purchaser.	68%	38	23
Sales of e-cigarette devices and refills from self-service vending machines should be banned.	89%	80	0
AoS restrictions should also apply to e-cigarette accessories.	69%	50	12
It should be an offence to proxy purchase e-cigarettes.	90%	80	1
**There Should be Regulation of Domestic Advertising.**	**89%**	**76**	**4**
Regulation of advertising of e-cigarettes should be in addition to that introduced by the TPD ^2^.	74%	63	4
Billboard advertising should be banned.	69%	56	6
Leafletting should be banned.	69%	54	8
Brand stretching should be banned.	69%	56	6
Free distribution of e-cigarettes should be banned.	70%	58	5
Nominal pricing for e-cigarettes should be banned.	70%	55	8
Point-of-sale advertising should be banned.	70%	49	14
Events sponsorship should be banned.	70%	58	5
There should be a Scottish Retailer Register for e-cigarette devices and refills.	74%	58	9
The offences and penalties regarding e-cigarettes should reflect those already in place for the Scottish Tobacco Retailers Register.	71%	56	8
**Use of E-Cigarettes in Enclosed Public Spaces Should be Banned.**	**72%**	**54**	**11**
There should be an age verification policy, “Challenge 25”.	74%	63	4
Penalties for selling e-cigarettes to under-18s should be the same as for tobacco.	66%	54	5
Sales of e-cigarettes by those under 18 should be prohibited.	70%	10	53

^1^ Response rate = percentage of actors that expressed a view either in support of or opposition to the proposed regulation. Balance represents missing or unclear data, i.e., actors not expressing a clear view. ^2^ TPD = Tobacco Products Directive. Bold: Elements of regulation that were most prominently discussed in the policy debates.

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
