# Peer review of "E-Cigarettes: A Disruptive Technology? An Analysis of Health Actors’ Positions on E-Cigarette Regulation in Scotland"

_ijerph, 2019, doi:10.3390/ijerph16173103_

Round 1

Reviewer 1 Report

Thank you for the opportunity to review manuscript ijerph-572777. This paper was a pleasure to read! I have only very minor suggestions:

I suggest adding a couple of sentences to the discussion (perhaps the future directions section?) noting that research is needed that examines how the e-cigarette debate is framed among those with a conflict of interest (i.e., academics, health professionals etc who have received vaping industry funding). I think it is important to acknowledge that although these individuals are “health professionals”, they have a vested interest in the debate.

I believe the findings of the paper are important globally too, not just in the UK (and Scotland specifically). Perhaps a sentence on Page 2 around Line 29 that notes the results have the potential to assist those working to improve public health in countries around the world?

Page 16 Lines 45-46: There seems to be a typographical error here?

Author Response

We have added a respective sentence to the discussion: “In order to shed light on any vested interests among health-focused actors and the potential impact such interests have on health policy debates, it would be valuable to examine conflicts of interests among actors who engage in e-cigarette debates and investigate how health-focused actors with a conflict of interest (e.g. academics, actors who receive commercial sector funding, pharmacists and other health professionals whose profits at least partly depend on e-cigarette sales, etc.) frame the debates.”

We have inserted a reference to those working to improve public health in countries around the world to the sentence on page 2, line 29.

We have deleted a word on page 16, line 46.

Reviewer 2 Report

The paper is well written and an excellent example of how to study disagreements among scientists, regulators, and industry.  My only comment pertains to two issues that were not covered that I believe are important and relevant.

Several recent articles have criticized the UK Royal College of Physicians, Royal Society of Public Health and the NHS for the often quoted '95% safer' report.  As we have now learned, the report was not founded on empirical evidence, and there may have been financial conflicts with committee members issuing the report. It should be noted in the introduction or discussion of the paper the impact this report had on expert and public opinion.  One could go as far as to say the report may have caused the regulatory disagreements and in fact lines up up nicely with tobacco industry Project Sunrise.

Second, the EU's e-cigarette directive calls for regulation of nicotine content at 20 mg/ml for unsupervised purchase.  This regulation has strong impact on the addictiveness of current e-cigarette devices such as JUUL and can explain differences in youth e-cigarette prevalence between the US, with no nicotine concentration limits, and several EU countries.  While this regulation is not a topic investigated by the study, it merits comment so that readers understand more broadly the regulatory context of Scotland, UK, England and other nations.           

Author Response

We now make reference to the Public Health England report in the discussion.

We refer to differences in US and UK debates on nicotine content regulation in order to highlight that conclusions about other political contexts must remain tentative.

We also receive the following comment by reviewer 3 by Miki Shen via e-mail:

"The authors should mention the fact that the U.S. and other countries’ failure to conduct clinical trials exavebates the “knowledge gap.”"

We now refer to the lack of rigorous clinical trials on page 15.